# Decrease in Myelin-Associated Lipids Precedes Neuronal Loss and Glial Activation in the CNS of the Sandhoff Mouse as Determined by Metabolomics

**DOI:** 10.3390/metabo11010018

**Published:** 2020-12-30

**Authors:** Emmanuelle Lecommandeur, Maria Begoña Cachón-González, Susannah Boddie, Ben D. McNally, Andrew W. Nicholls, Timothy M. Cox, Julian L. Griffin

**Affiliations:** 1Department of Biochemistry and Cambridge Systems Biology Centre, University of Cambridge, Cambridge CB2 1GA, UK; emmanuelle.lecommandeur@gmail.com (E.L.); boddiesusannah@gmail.com (S.B.); mcnallyb14@gmail.com (B.D.M.); 2Department of Medicine, Cambridge Biomedical Campus, Cambridge CB2 0QQ, UK; mcb23@medschl.cam.ac.uk (M.B.C.-G.); tmc12@medschl.cam.ac.uk (T.M.C.); 3GlaxoSmithKline, Stevenage SG1 2NY, UK; andrew.w.nicholls@gsk.com; 4Hammersmith Campus, UK Dementia Research Institute at Imperial College, Burlington Danes Building, Imperial College London, Du Cane Road, London W12 0NN, UK; 5Section of Biomolecular Medicine, Department of Metabolism, Division of Systems Medicine, Digestion and Reproduction, The Sir Alexander Fleming Building, Exhibition Road, South Kensington, Imperial College London, London SW7 2AZ, UK

**Keywords:** lipidomics, metabolomics, lysosomal disorders, β-hexosaminidase, galactosylceramides, bis(monoacylglycero)phosphates, plasmalogens

## Abstract

Sandhoff disease (SD) is a lysosomal disease caused by mutations in the gene coding for the β subunit of β-hexosaminidase, leading to deficiency in the enzymes β-hexosaminidase (HEX) A and B. SD is characterised by an accumulation of gangliosides and related glycolipids, mainly in the central nervous system, and progressive neurodegeneration. The underlying cellular mechanisms leading to neurodegeneration and the contribution of inflammation in SD remain undefined. The aim of the present study was to measure global changes in metabolism over time that might reveal novel molecular pathways of disease. We used liquid chromatography-mass spectrometry and ^1^H Nuclear Magnetic Resonance spectroscopy to profile intact lipids and aqueous metabolites, respectively. We examined spinal cord and cerebrum from healthy and *Hexb*^−/−^ mice, a mouse model of SD, at ages one, two, three and four months. We report decreased concentrations in lipids typical of the myelin sheath, galactosylceramides and plasmalogen-phosphatidylethanolamines, suggesting that reduced synthesis of myelin lipids is an early event in the development of disease pathology. Reduction in neuronal density is progressive, as demonstrated by decreased concentrations of *N*-acetylaspartate and amino acid neurotransmitters. Finally, microglial activation, indicated by increased amounts of myo-inositol correlates closely with the late symptomatic phases of the disease.

## 1. Introduction

Sandhoff disease (SD) is a GM2 gangliosidosis (in the abbreviation GM2, G refers to ganglioside, the M is for monosialic, and 2 refers to the fact that it was the second monosialic ganglioside discovered) caused by mutations in the gene encoding the β subunit of β-hexosaminidase (HEXB), leading to a deficiency in the lysosomal enzymes β-hexosaminidase A (HEX A) and B (HEX B) [1]. HEX A and HEX B cleave *N*-acetylglucosamine (GlcNAc) and *N*-acetylgalactosamine (GalNAc) residues from a variety of substrates, including ganglioside GM2, abundant in nervous tissue. Over 50 different disease-causing mutations in *HEXB* have been reported [2], giving rise to a wide spectrum of disease onset and symptoms. Typically, in the infantile form, motor weakness is detected between the ages of three and six months, followed by severe hypotonia and blindness, with death occurring between two and four years. In the juvenile and adult forms of the disease, the clinical manifestations and progression are more heterogeneous [1,3,4]. Accumulation of gangliosides GM2 and GA2 (asialylated GM2) is detected throughout the central nervous system (CNS). Furthermore, the unacylated ganglioside, lyso-GM2 has been reported to be increased in the brain and plasma of SD patients and in the mouse model [5].

The mouse model of SD, *Hexb^−/−^*, exhibits manifestations similar to those of human patients albeit on a shorter time scale, with a lifespan of four to six months [6,7]. Deterioration of motor function is first detected at three months of age and by five months there is an almost complete lack of hind limb movement accompanied by muscle atrophy. Pathological storage material, gangliosides as well as glycosaminoglycans, are detectable in all regions of the CNS. Importantly, no storage material is detectable in heterozygous human carriers of SD, nor in heterozygote mice (*Hexb^+/−^*) [6]. Apoptotic neurons have been detected in the murine brain and spinal cord at the peri-symptomatic age of three months. Similar features were observed in autopsy samples from brain and spinal cord from a 19-month-old SD patient [7]. The CNS expression of genes known to be elevated during macrophage/microglia-mediated inflammatory response has been reported in brains of SD patients [8]. In the SD mouse, overexpression of inflammatory genes has been detected commencing at age one month, with inflammation and microglial activation becoming more widely distributed with time [8]. However, a rigorous dissection of the biochemical pathways contributing to the evolution of SD has not been undertaken. In this work, we sought to identify metabolic changes associated with SD progression and pre-symptomatic alterations involved in the pathogenesis using the mouse as an authentic model of disease.

## 2. Results

### 2.1. LC-MS and NMR Spectroscopy Show Metabolic Changes Directly Resulting from the Enzyme Deficiency

Analysis of cerebrum and spinal cord separated the spectra according to genotype using multivariate analysis (partial least squares discriminate analysis (PLS-DA)) of the LC-MS dataset at each of the four time points studied. GM2 affects all the CNS but the initial microscopic neuropathology is in the cerebrum which is neurone rich in the cortex (grey matter), and also clinically associated with the rapid dementia. On the other hand, with subcortical axons, the spinal cord is rich in myelinated axons. Thus, these major structures typify the range of “grey-white” tissue divisions of the CNS studied in this disease.

Increases in the concentration of several species of gangliosides GA2 and GM2 were detected using negative ionisation mode LC-MS and were highly discriminatory between controls and *Hexb ^−/−^* mice (Figure 1A). GA2 (d18:1/18:0) was the most increased species in cerebrum and spinal cord tissue from SD mice, being present only at trace concentrations in controls (Figure 1B,C; Appendix A). The increase in total GA2 and GM2 ganglioside content peaked at four months (655-fold increase in cerebrum tissue and 232-fold increase in spinal cord tissue from *Hexb ^−/−^* mice compared to controls).

Alongside GA2 and GM2 gangliosides, bis(monoacylglycero)phosphates (BMPs) were the most discriminatory lipids detected in negative ion mode LC-MS between controls and *Hexb ^−/−^* mice. BMPs are lipids that localise to the inner membrane of endosomes and lysosomes [9]. Seven BMP species accumulated in the CNS of *Hexb ^−/−^* mice over time, with BMP (22:6/22:6) being the first species to be significantly increased in one-month-old SD mice compared to controls, and by far the more abundant at any time point in SD mice (18-fold increase in the brain and 26-fold increase in spinal cord from four-month-old SD mice compared to controls). At the last time point measured, the concentrations of all the BMPs detected in SD spinal cord and cerebrum were significantly increased (Figure 1D,E).

Aqueous metabolites were analysed by high resolution ^1^H-NMR spectroscopy. Many of the metabolites detected by ^1^H NMR spectroscopy in brain tissue extracts can also be detected in ^1^H Magnetic resonance spectroscopy (MRS) studies of the intact brain, and thus, there is a wide literature related to brain neurodegeneration we could compare our results with. The presence of an extra resonance at 2.07 ppm, corresponding to *N*-acetylhexosamine (HexNAc), was a unique and striking feature in SD mice, detected at every time point studied, and in both cerebrum and spinal cord (Figure 2). This finding confirms previous observations in SD patients and mice [10,11]. Moreover, the corresponding resonance increased as the disease progressed from one to four months (Figure 2), indicating that the storage material increases with time.

### 2.2. Alterations in Lysophosphatidylcholines and Lipid Components of Myelin Are Early Stage Changes Associated with Disease Progression

Owing to evidence of the role of lysophosphatidylcholines (LPCs) in β-amyloid-induced neuronal loss in vitro [12], we studied the metabolism of lysophospholipids using open profiling LC-MS. Several LPC species were affected in *Hexb ^−/−^* mice; the concentration of LPC 18:0 was decreased in cerebrum tissue from *Hexb ^−/−^* mice at one month, whereas the concentration of LPC 16:0 was increased at two months and the concentration of LPC 18:1 was increased at two and three months (data not shown). Alongside these changes, a time-dependent increase in the concentrations of LPC 20:4 and LPC 22:6 was detected in SD cerebrum. This increase was significant from the earliest time point studied (Figure 3A). However, the increase in these two species was not significant at four months due to larger intra-group variations. In spinal cord tissue, the same trend was observed, although the increase in LPC 20:4 and LPC 22:6 was not significant at one month (Figure 3B). In addition, there was a time-dependent increase in the concentration of LPC 18:1 in spinal cord tissue from *Hexb ^−/−^* mice.

Myelin, the insulating axonal sheath that ensures correct transmission of nerve impulses, has a high lipid content (about 70% of its dry weight in humans and rats) [13]. Galactosylceramides (GalCers) and glucosylceramides (GluCers), also called cerebrosides, are the most typical of these lipids, followed by ethanolamine-containing plasmalogens [13]. In the present study, our LC-MS/MS method could not discriminate between GalCers and GluCers, and so we refer to the detected species as hexosylceramides (HexCer). HexCer species were measured using positive ion mode LC-MS and identified using their exact mass and the presence of an intense ion *m*/*z* 264.3, corresponding to the loss of the acyl group bound to the amine of the ceramide and to a double dehydration of the ceramide moiety, in the MS/MS mass spectrum. Plasmalogen-phosphoethanolamine (PE) and HexCer species were decreased in the cerebrum of one-month-old SD mice (Figure 4A). The targeted measurement of plasmalogen-PE (p36:1) in cerebrum and spinal cord confirmed that its concentration was decreased in SD mice compared to controls at one and two months (Figure 4B).

The analysis of CNS tissues using ^1^H-NMR spectroscopy revealed large metabolic alterations between control and SD mice (Figure 5A). *N*-acetylaspartate is a marker of neuroaxonal tissue throughout the brain and the spinal cord, measurable using NMR spectroscopy [10,11,14,15,16,17]. Multivariate analysis revealed that the most discriminatory decrease detected by ^1^H NMR spectroscopy in CNS tissue from SD mice was a reduction in the concentration of *N*-acetylaspartate (Figure 5A). It was decreased in the spinal cord of SD mice at all time points studied (one to four months) with a severe drop starting at the peri-symptomatic age of three months (Figure 5B). In the cerebrum, the decrease in *N*-acetylaspartate reached significant values only in age groups one and two months, intra-group variations being larger than in the spinal cord (Figure 5D). At the symptomatic humane end point of four months, the concentration of *N*-acetylaspartate was reduced by 13.7% (although non-significant, *p* = 0.18; for resonances at 2.70–2.72 ppm) and 46.5% (*p* = 0.0003) in cerebrum and spinal cord, respectively.

Glutamate, aspartate and γ-aminobutyric acid (GABA) were quantified using ^1^H-NMR spectroscopy in CNS tissue (Figure 6). The concentration of glutamate was reduced in the cerebrum of two-month-old SD mice and from two months onwards in spinal cord tissue (Figure 5G). The concentration of aspartate was also significantly decreased in cerebrum tissue from one and four-month-old *Hexb ^−/−^* mice and in the spinal cord at three and four months (Figure 5H). Finally, the concentration of GABA was decreased in the cerebrum of the mouse model of SD at one and four months and in spinal cord tissue at two and four months (Figure 5G,H). The decreases in neurotransmitter concentrations were highly discriminatory between both groups studied for CNS tissues (Figure 5A).

Myo-inositol is a glia marker, more specifically thought to be a marker for astrogliosis, [10,11,18] that can be detected using ^1^H-NMR spectroscopy. At four months, an increase in the concentration of myo-inositol was among the most discriminatory metabolic changes detected in SD cerebrum and spinal cord (Figure 5A). Surprisingly, the concentration of myo-inositol in SD spinal cord at ages one and two months was significantly decreased compared to healthy controls, though an increase was clearly observable from three months onwards, correlating with the symptomatic phase of the disease (Figure 5C). Moreover, the calculation of the *N*-acetylaspartate to myo-inositol ratio, representative of the proportion of neuronal cells to glial cells, demonstrated a time-dependent decrease of neuronal density in favour of the microglial population in SD CNS (data not shown).

The branched-chain amino acids valine and leucine were detected using ^1^H-NMR spectroscopy. In cerebrum, their concentrations were increased in one month SD mice compared to controls (Figure 5E). In spinal cord, the concentration of leucine was increased at one (Figure 5F) and two months in the SD mouse compared to controls, while the concentration of valine was increased at one, three and four months.

### 2.3. Sphingoid Bases Accumulate in the SD Mouse Brain

To investigate sphingolipid metabolism further we performed a separate tissue extraction and chromatography to analyse sphingoid bases in the tissue (Figure 7). C16 species showed limited variance between genotypes with no significant alterations in any age group. However, C18 species appear to be key features of disease pathology. C18 sphinganine is significantly increased in 1 month mice (*t*-test, *p* = 0.0041) but this increase is not maintained at a significant level in samples from older mice. This suggests an upregulation of de novo sphinganine synthesis or decline in ceramide synthesis downstream in 1 month samples. On the other hand, C18 sphingosine accumulation in SD samples is observed across all age groups, with significant results in 1 and 3 month old mice (*t*-test, *p* = 0.044, *p* = 0.025, respectively). Impairment of the salvage pathway should reduce the production of sphingosine, so this suggests that compensatory upregulation of sphingosine synthesis may occur in SD.

## 3. Discussion

We describe here the course of SD at a metabolic level, with numerous abnormalities in diverse metabolites affected in the mouse model of SD (Figure 8).

An increase in GM2 was observed at every time point studied in nervous tissues from the *Hexb ^−/−^* mice. The concentration of various species of the asialo-sphingolipid, GA2, steadily increased in the CNS as the disease progressed, highlighting the extent of the lysosomal storage due to deficiency in HEX A and B in the mouse model of SD. The increases in BMPs, lipids located to the inner membranes of lysosomes and late endosomes [9] most probably reflect the impairment of the endo-lysosomal degradation pathway, and the accumulation of these organelles. Similar increases have been observed previously in several lysosomal disorders including Niemann–Pick diseases, neuronal ceroid lipofuscinoses, SD and also drug-induced phospholipidosis [19,20,21,22,23,24,25,26,27]. In addition, BMPs have been reported to stimulate the degradation of glycosphingolipids in lysosomes [9] and participate in the transport of cellular cholesterol and its degradation [9,28,29]. Therefore, increased concentrations of BMPs could also denote an adaptive mechanism for the cells to degrade lipids accumulating due to the enzyme deficiency in SD. In our study, BMP (22:6/22:6) was the most abundant BMP species in the CNS from SD mice, similar to what was observed in the brain of mouse models of neuronal ceroid lipofuscinosis [19]. The brain is particularly rich in docosahexaenoic acid (DHA, C22:6 (n − 3)), and the accumulation of BMP (22:6/22:6) probably reflects the high proportion of this fatty acid in cerebrosides. In addition, the ^1^H-NMR spectroscopy demonstrated an accumulation of HexNAc as a result of the deficiency in β-hexosaminidase in the mouse model of SD. This confirms previous data obtained using magnetic resonance spectroscopy on patients and on a mouse model of SD [10,11].

At one month, decreases in the main myelin lipid components, plasmalogen-PE and HexCer, were detected in the CNS of SD mice. This agreed with a previous report showing a significant reduction in the gene expression of UDP-galactose ceramide galactosyltransferase (*Cgt*), a key-enzyme in the biosynthesis of myelin lipids such as GalCer [30] from eight weeks of age, and decrease in structural protein components of myelin from five weeks of age [31]. Indeed, it is well documented that myelin-associated lipids are decreased in the brains of humans and animal models of SD [32,33]. Unexpectedly, the concentration of plasmalogen-PE (p36:1), a major component of myelin, was not significantly different between SD mice and controls at three and four months of age despite clear differences detected at one and two months. In mice, biosynthesis of GalCer principally occurs between postnatal days 10 to 20, whereas myelin formation is maximal between days 20 to 25 [34,35]. One possibility is that the decreased abundance of myelin lipid components, clearly significant at the earliest time points, reflects impaired biosynthesis of the lipid components of myelin rather than myelin destruction. However, these changes could also be due to a reduction in the oligodendrocyte population or impaired metabolism in this cell type; therefore, further study of the cellular changes within the brain would be necessary to understand these metabolic changes. As the deficiency of the myelin lipids occurs before physical manifestations can be detected in SD mice, defects in myelination in SD mice probably should not be attributed solely to neuronal pathology, which, as illustrated by the continuous decrease in *N*-acetylaspartate, becomes more prominent with time. Lysophosphatidylcholines have previously been shown to induce demyelination [36,37,38,39]. Hence, early increases in the concentrations of LPC species in the nervous system of SD may contribute to the impairment of myelination occurring at the early stages of the disease.

The concentration of myo-inositol was increased in the spinal cord of four-month-old *Hexb*
^−/−^ mice compared to controls, presumably reflecting the activation of glial cells in the CNS as part of an inflammatory process as previously observed in models of neurodegeneration [18]. These results agree with previous reports showing that microglial activation and production of inflammatory molecules were increased in an age-dependent manner in SD mice [8,40]. Moreover, our results agree with previous findings suggesting that the progressive activation and expansion of microglia occurs concomitant with neurodegeneration in SD and may trigger loss of neurons [8,41]. Indeed, we demonstrate that the concentrations of *N*-acetylaspartate and myo-inositol follow an inverse temporal trend, suggesting that both processes are interrelated.

A notable finding in this study was the time-dependent increase in the concentration of several lysophospholipid species in the brain and spinal cord in SD mice. Peripheral blood monocytes participate in the expansion of microglial population in SD [8,41] and the chemoattractant properties of LPCs, reported in the brain of the mouse model of infantile neuronal ceroid lipofuscinosis, appear to be responsible for infiltration by phagocytes [42]; it is possible that these bioactive lipids have similar role in the neuropathology of SD. In Alzheimer’s disease, increased activity of cytosolic phospholipase A2 leads to elevated LPC concentrations which also stimulate release of pro-inflammatory cytokines [43]. Thus, we propose that the increased LPCs observed in CNS tissues obtained from mice with SD may play a similar role, recruiting blood monocytes to expand the microglial population in response to the accumulation of gangliosides, creating an inflammatory state that contributes to neuronal loss, as suggested in previous studies [8,41].

Elevated levels of C18 sphingosine were observed across all age groups. This species is a competitive inhibitor of protein kinase C (PKC) activation, a crucial regulator of several processes in the brain [44]. PKC regulates the cytoskeleton dynamics required for synapse formation and maintenance, and also regulates neurotransmission and synaptic plasticity by phosphorylating transporters, ion channels and G protein-coupled receptors [45]. Therefore, this alteration may contribute to the impaired brain development that causes mental retardation in SD patients.

Finally, the concentrations of three neurotransmitters, aspartate, glutamate and GABA, were decreased in the cerebrum and spinal cord of the mouse model of SD at most time points. This might be related to the decrease in neuronal density as evidenced by a reduction in *N*-acetylaspartate, but it more likely represents a combination of impaired neuronal function and cell loss, as seen from previous studies [46].

One limitation of the current study is that the “open profiling” method that was used for lipidomics analyses the high concentration lipids in tissue extracts that are readily soluble in chloroform/methanol. However, it may not detect low abundance lipids, or lipid classes that are poorly soluble in chloroform. One notable lipid class we failed to detect using this method is the sulfatides, an important class of lipids associated with myelin and known to be altered in SD. Furthermore, the suggested neurotoxic lyso-GM2 and lyso-GA2 lipids were not detectable using this assay. More dedicated assays could be used to follow these changes in the future.

Although the role of demyelination in SD requires further investigation, its early occurrence indicates that it contributes importantly to pathogenesis and represents a point of no return of the neurological disease following therapeutic gene transfer, as previously indicated by histopathological studies [31]. If the critical role of demyelination is confirmed, delivering corrective treatments such as gene therapy before injury to myelin is established would be essential in order to achieve optimal therapeutic effects. Indeed, gene therapy-based approaches have corrected myelin deficiencies not only for GM2 storage in SD but also for GM1 storage in GM1 gangliosidosis [32,47].

## 4. Materials and Methods

Mouse model of Sandhoff disease: The SD mouse model (strain: B6; 129S-Hexb^tm1Rlp^), developed by disruption of the *Hexb* gene [6], was obtained from the Jackson Laboratory, USA. Homozygous males were mated with heterozygous females, or heterozygous animals were interbred, to maintain the colony. Mice were fed ad libitum and killed by asphyxiation at one, two, three or four months of age (*n* = 5 per age and genotype). Tissues were dissected and frozen immediately on dry ice before storage at −80 °C. All studies were conducted under Home Office licence and conformed to the UK Animals (Scientific Procedures) Act 1986.

Metabolite extraction: Metabolites were extracted from 50 mg of cerebrum and spinal cord tissue using the protocol from Le Belle and co-workers [48] and briefly described below. First, 6 mL methanol and chloroform in a 2:1 ratio (v/v) was used per gram of tissue. The mixture was homogenised in a TissueLyser (Qiagen, Germany) and sonicated for 15 min at room temperature (22 °C). To the mixture was added 2 mL chloroform and 2 mL water per gram of tissue to form an emulsion. Then, the samples were centrifuged for 20 min at 20,844× *g* at 22 °C in a centrifuge 5424 (Eppendorf) to generate distinct organic and aqueous fractions and a protein pellet. The organic and aqueous layers were transferred to separate tubes. The entire process was repeated on the protein pellet and remnants of the aqueous and organic fractions in order to perform a double extraction to maximize extraction efficiency. Fractions separated during the second extraction were combined with the first ones. Finally, aqueous extracts were dried overnight in a Concentrator Plus (Eppendorf, Germany) evacuated centrifuge at room temperature (22 °C) while organic layers were dried in a fume hood under a stream of nitrogen gas. All dried fractions were stored at −80 °C until analysis.

Profiling of intact lipids: The open profiling of lipids was performed using the dried lipid fraction obtained after metabolite extraction (10 mg tissue equivalent). Chromatography was performed on a Acquity UPLC^®^ system using a Acquity UPLC^®^ Charged Surface Hybrid (CSH) C18 column (1.7 µm by 2.1 mm by 100 mm) (Waters Inc., Milford, MA, USA) at 55 °C. Lipid fractions were diluted in isopropanol/acetonitrile/water (2:1:1 v/v/v). The injection volume was 2 µL. The flow rate was 0.4 mL/min. Mobile phase A was composed of 60% acetonitrile and 40% water with the addition of 10 mM ammonium formate. Mobile phase B was 10% acetonitrile and 90% isopropanol with 10 mM ammonium formate. The chromatographic gradient progressed from 40% mobile phase B up to 99% B over 18 min, followed by 2 min of equilibration at 40% B.

Mass spectrometry was performed on a Xevo G2 quadrupole-time of flight (Q-ToF) mass spectrometer (Waters Inc., Milford, MA, USA) in positive and negative ion modes using a scan time of 0.2 s, a collision energy of 6 V for each single scan, and a collision ramp from 25 to 40 V for the fragmentation function in positive ion mode and from 30 to 50 V in negative ion mode. The capillary voltage was 2 kV, sampling cone was 30 V, extraction cone was 3.5 V and source temperature was 120 °C. Leucine enkephalin was used as a lockmass to improve mass accuracy throughout the analysis, and 5 mM sodium formate was used to calibrate the instrument prior to analysis (the maximum threshold for mass difference between the measured mass and the exact mass was set at 5 ppm). Chromatograms were processed using the MarkerLynx XS tool from the MassLynx software (version 4.1, Waters Inc., USA) using peak detection analysis with the following parameters: Retention time: 0 to 18 min, mass range: 50 to 2300 Da and tolerance for the mass value: 0.05 Da, minimum intensity to consider for a spectral peak: 2000 counts. Data were de-isotoped and metabolites were identified using the following online databases: www.lipidmaps.org (Lipid Metabolites and Pathways Strategies), www.hmdb.ca (Human Metabolome Database) and www.genome.jp/kegg/ (the Kyoto Encyclopaedia of Genes and Genomes). The abundance of specific species was calculated as the average area under the curve of the LC-MS chromatogram for each corresponding *m*/*z*.

Targeted analysis of ceramides: Targeted analysis was performed using a 4000 QTRAP^®^ triple quadrupole instrument operated in positive ion mode. The parameters for each mass transition were optimised individually by injecting standards. Analytes were monitored in multiple reaction monitoring (MRM) mode using the following parameters in Table 1.

Data were processed using the Analyst software (version 1.6; AB Sciex, Darmstadt, Germany). The abundance of specific species was calculated as the average area under the curve of the LC-MS chromatogram for each corresponding *m*/*z*.

Analysis of aqueous metabolites by ^1^H-NMR spectroscopy: The dried aqueous samples were dissolved in 600 μL of a 0.5 mM solution of sodium-3-(trimethylsilyl)-2, 2, 3, 3-tetradeuteriopropionate (TSP) in deuterium oxide (D_2_O) containing 0.1% sodium azide to protect the sample from bacterial contamination. The TSP served as an internal chemical shift reference (^1^H δ0.0) and the spectrometer field frequency lock was provided by D_2_O. The sample was transferred to a 5 mm NMR glass tube and analysed on a 11.74 T (500 MHz ^1^H observation frequency) NMR spectrometer using a 5 mm ^1^H{^13^C}-TXI NMR probe fitted with automatic tuning and matching (ATMA) (Bruker, Karlsruhr, Germany). ^1^H NMR spectra were obtained using the NOESY-presaturation pulse sequence, based on the first increment of a 2D nuclear Overhauser effect spectroscopy (NOESY) pulse sequence to suppress the residual ^1^H water signal. NMR spectra were acquired using 128 scans collected into 16 k data points with an acquisition time of 4.09 s, a relaxation time of 2 s and a mixing time of 150 ms. Solvent presaturation was applied during the relaxation and mixing times and the total pulse recycle time was 2 s. All data were collected with the sample temperature set to 27 °C.

NMR spectral assignment was achieved with reference to previous literature, online databases (www.hmdb.ca, Human Metabolome Database, version 3.5) and the Chenomx spectral database contained in Chenomx NMR Suite (version 7.5, Chenomx, Edmonton, Canada). NMR spectra were processed in ACD NMR Manager (version 12, Advanced Chemistry Development, Toronto, Canada). Fourier transformation was applied to the spectra and TSP was set as a reference at 0.0 ppm. Phasing and baseline correction were carried out manually. The region corresponding to water was deleted (δ4.5–5.0 ppm) and remaining spectral regions were integrated from 0.5 to 9.5 ppm. Resonances having the least overlap with peaks corresponding to other metabolites were chosen to be integrated for the quantification of each metabolite.

### Sphingoid Base Analysis by LC-MS

Lipids were extracted following the method proposed by Sullards and co-workers [49]. Approximately 10 mg of tissue was weighed for extraction. Then, 90 μL of PBS and 750 μL of methanol/chloroform (2:1) was added to each sample, prior to lysis in a TissueLyser (Qiagen) for 12 min, at 16 Hz. The homogenates were incubated overnight at 48 °C. After cooling, 75 μL potassium hydroxide (0.1 M in methanol) was added, and the samples were sonicated for 10 min. Another incubation at 37 °C for 2 h completed the extraction. 0.4 mL of each sample extract was decanted for reverse phase LC-MS and centrifuged (16, 100 g, 5 min) to remove insoluble material. An Avanti internal standard cocktail (Cer/Sph Mixture 1, LM6002-1EA) was added to each sample (10 μL) to enable quantification (including d17:1 sphingosine and d17:0 sphinganine), and the sample was dried under nitrogen flow. 

Dried extracts were reconstituted in 100 μL of the appropriate mobile phase (reverse phase solution A/reverse phase solution B (60:40)), sonicated for 10 min and centrifuged (16, 100 g, 10 min). An adapted version of the method proposed by Shaner and colleagues [50] was used. Then, 10 µL of sample was injected onto a C18 CSH column, 2.1 by 50 mm (1.7 µm pore size; Waters), which was held at 55 °C using an Ultimate 3000 UHPLC system (Thermo Fisher Scientific, Hemel Hempstead, UK). The mobile phase, comprised of solvents A (CH_3_OH/H_2_O/CHCOOH; 58:41:1) and B (CH_3_OH/CHCOOH; 99:1), was run through the column in a gradient (40% B, increased over a gradient to 100% B after 0.5 min, held at 100% B from 2.3 min and then returned to 40% at 7.6 min for the final 0.5 min). Total run time was 8.1 min, with a flow rate of 0.6 mL/min. 5 mM ammonium formate was used as an additive in solvents A and B. 

A LTQ Orbitrap Elite Mass Spectrometer (Thermo Fisher Scientific) was used with a heated electrospray, using a source temperature of 400 °C and a capillary temperature of 380°C. In positive mode, a 3.5 kV spray voltage was used, while in negative it was 2.5 kV. Data was collected using the Fourier transform mass spectrometer (FTMS) analyser. The resolution was set to 60,000 and the data was obtained in profile mode. The full scan was performed across an *m*/*z* range of 110–2000. 

Statistical analysis: Multivariate data analysis was performed using SIMCA-P+ (versions 12.0 and 13.0, Umetrics AB, Sweden). Mass spectrometry data sets were unit variance scaled, while NMR spectroscopy data were Pareto scaled, prior to analysis. The pattern recognition methods used were principal components analysis (PCA) to profile the dominant variation in the dataset and PLS-DA to maximise classification of groups.

Univariate statistics were conducted using the GraphPad Prism package (version 4, GraphPad Software, San Diego, CA, USA). The two-way unpaired two-tailed Student’s *t*-test was used to compare *Hexb ^−/−^* mice with healthy controls. For reporting purposes, a significance value was set at *p* ≤ 0.05.

## Figures and Tables

**Figure 1 metabolites-11-00018-f001:**
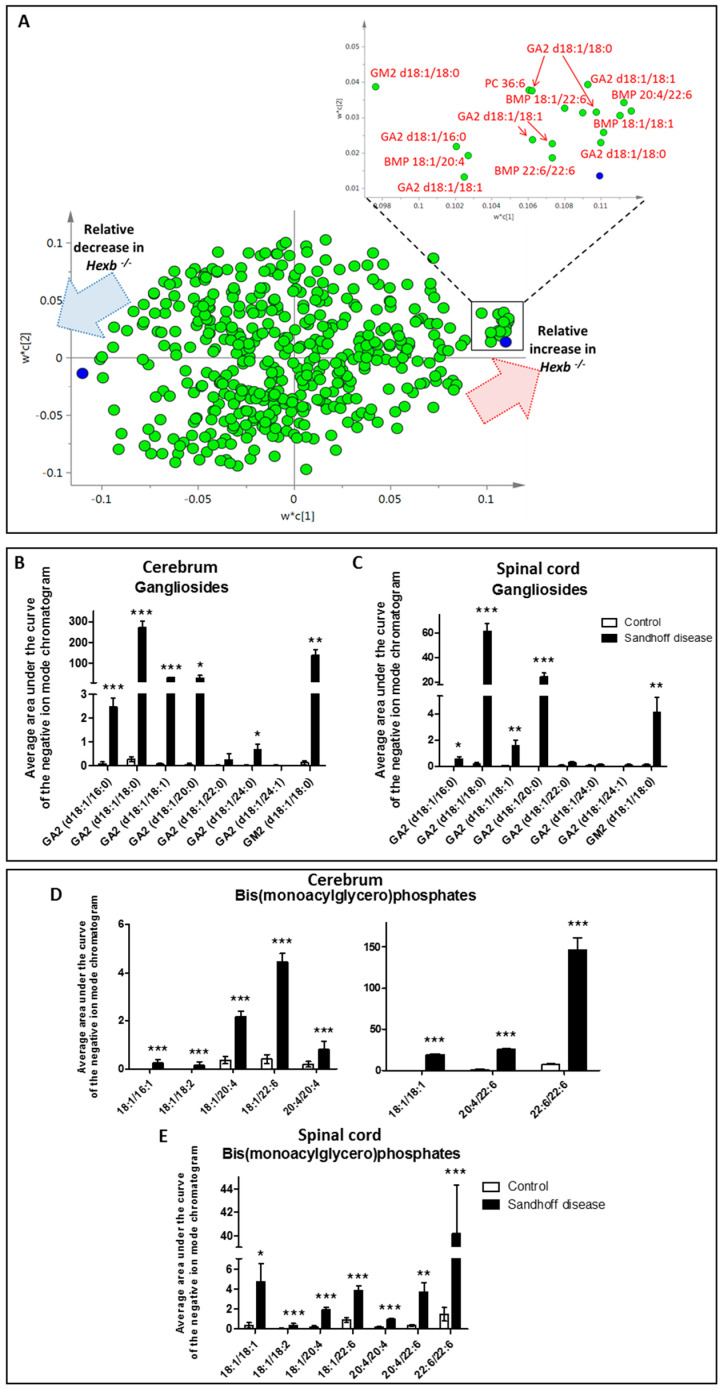
Metabolic variations between four-month-old *Hexb ^−/−^* mice and controls detected using open profiling negative ion mode LC-MS were dominated by increases in gangliosides and bis(monoacylglycero)phosphates (BMPs). (**A**) partial least squares discriminate analysis (PLS-DA) loading scatter plot from the analysis of negative mode LC-MS of cerebrum tissue from four-month-old control and *Hexb ^−/−^* mice. The most discriminatory increases in *Hexb ^−/−^* mice compared to controls are labelled in red. (**B**) concentration of ganglioside species in cerebrum tissue from four-month-old control and *Hexb ^−/−^* mice. (**C**) concentrations of ganglioside species in spinal cord tissue from four-month-old control and *Hexb ^−/−^* mice. (**D**) concentration of BMP species in cerebrum tissue from four-month-old control and *Hexb ^−/−^* mice. (**E**) BMP concentrations in spinal cord tissue from four-month-old control and *Hexb ^−/−^* mice. Results are presented as mean +/− SEM. Significance level quoted for Student’s *t*-test * *p* ≤ 0.05; ** *p* ≤ 0.01; *** *p* ≤ 0.001.

**Figure 2 metabolites-11-00018-f002:**
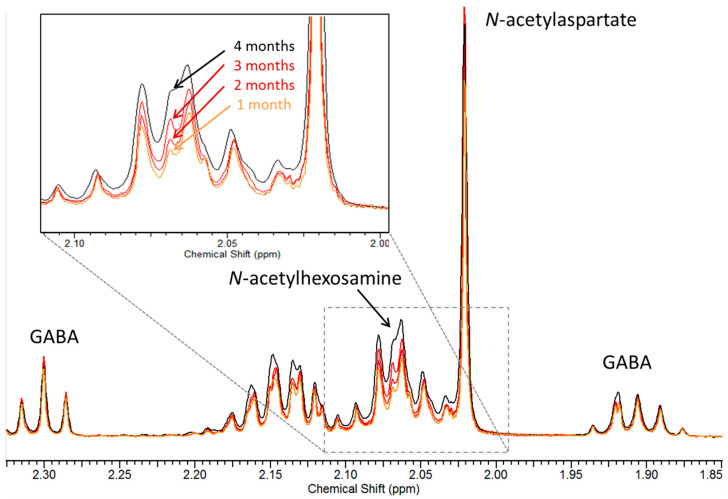
Expanded region (1.85–2.35 ppm) from the ^1^H-NMR spectra of cerebrum tissue from one, two, three and four-month-old *Hexb ^−/−^* mice. The resonance corresponding to *N*-acetylhexosamine, at 2.07 ppm, increased with time (insert). Other major resonances present in this region are as labelled. GABA: γ-aminobutyric acid.

**Figure 3 metabolites-11-00018-f003:**
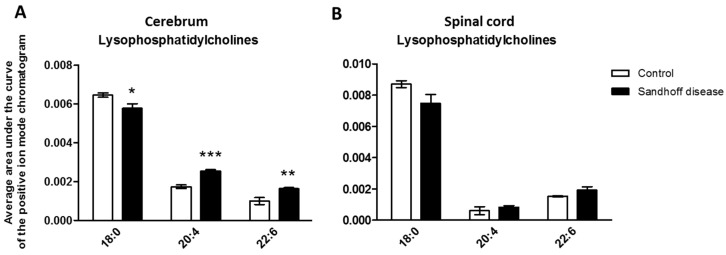
Concentration of three lysophosphatidylcholine species in cerebrum and spinal cord tissues from one-month-old control and *Hexb ^−/−^* mice. (**A**) concentration of lysophosphatidylcholine (LPC) 18:0, 20:4 and 22:6 in the cerebrum. (**B**) concentration of LPC 18:0, 20:4, and 22:6 in spinal cord. Results are presented as mean +/− SEM. Significance level quoted for Student’s *t*-test * *p* ≤ 0.05; ** *p* ≤ 0.01; *** *p* ≤ 0.001.

**Figure 4 metabolites-11-00018-f004:**
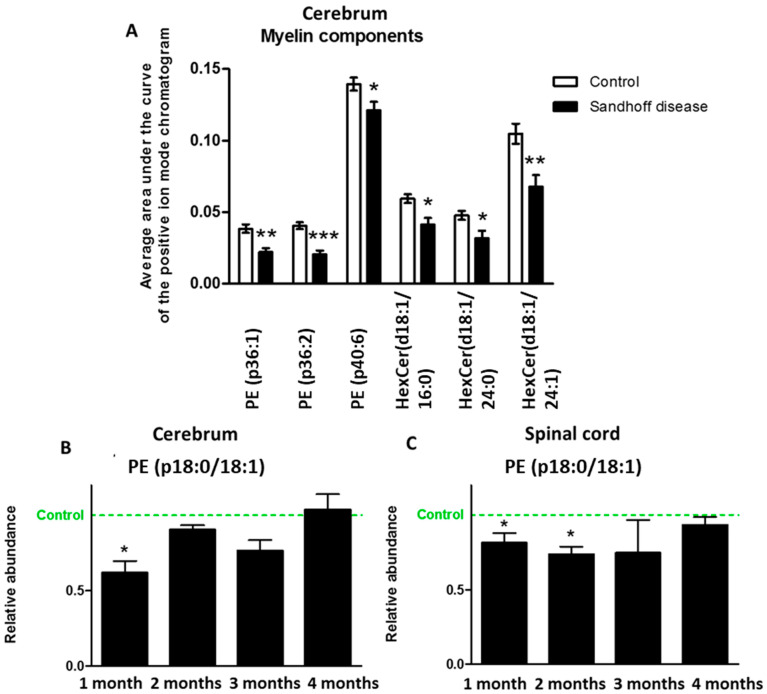
Measurement of myelin components using LC-MS. (**A**) concentration of major components of myelin, plasmalogen-phosphatidylethanolamine and hexose-ceramide species, in cerebrum tissue from one-month-old control and *Hexb ^−/−^* mice. (**B**) concentration of plasmalogen-phosphatidylethanolamine 36:1 (PE (p18:0/18:1)) in cerebrum from Sandhoff disease (SD) mice relative to controls (abundance set to 1). (**C**) concentration of plasmalogen-phosphatidylethanolamine 36:1 (PE (p18:0/18:1)) in spinal cord from SD mice relative to controls (abundance set to 1). Results are presented as mean +/- SEM. Significance level quoted for Student’s *t*-test * *p* ≤ 0.05; ** *p* ≤ 0.01; *** *p* ≤ 0.001.

**Figure 5 metabolites-11-00018-f005:**
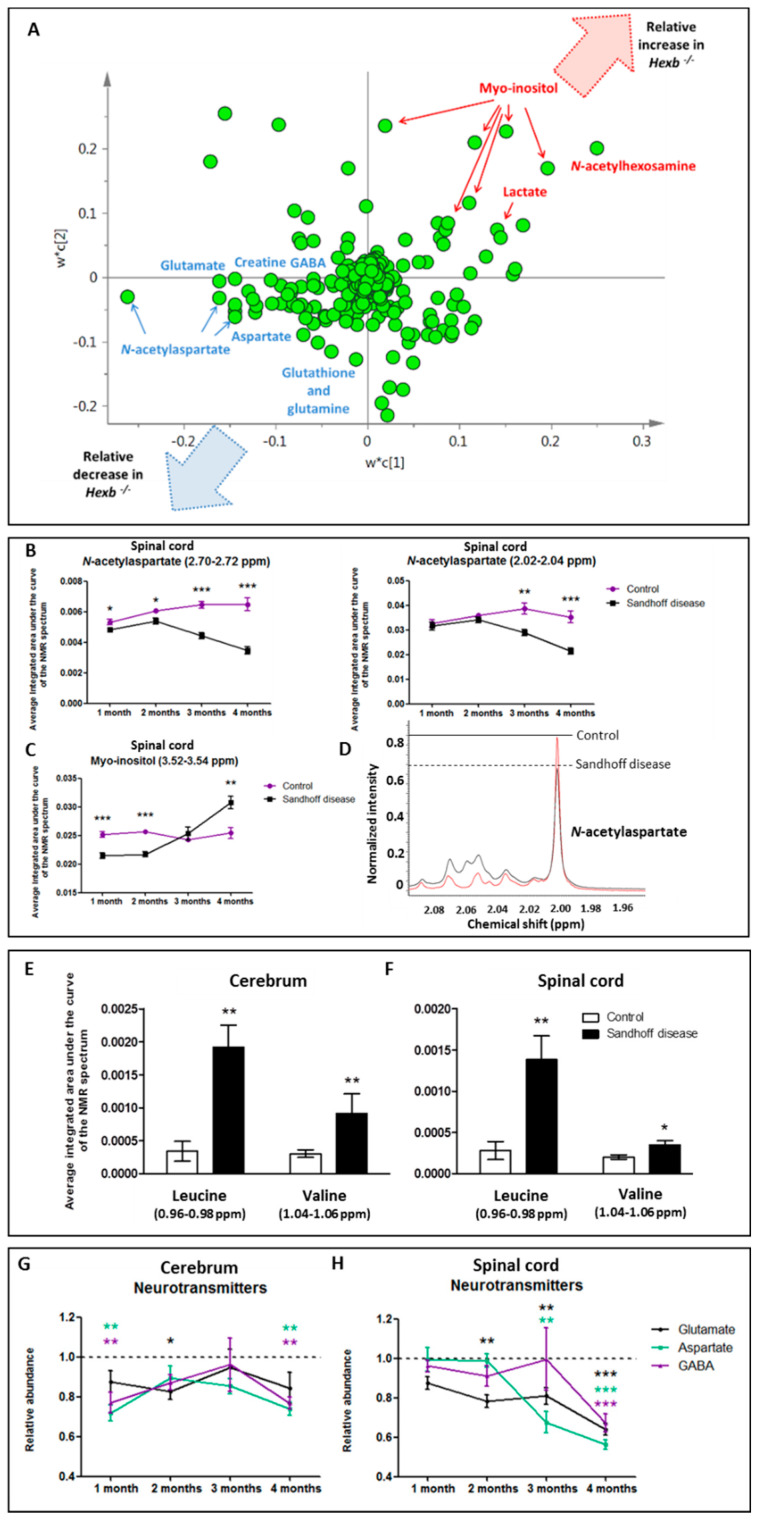
Metabolic variations between *Hexb ^−/−^* mice and controls from one to four months of age detected using NMR spectroscopy. (**A**) PLS-DA loading scatter plot from the analysis by ^1^H-NMR spectroscopy of spinal cord tissue from four-month-old control and *Hexb ^−/−^* mice. The most discriminatory decreases in *Hexb ^−/−^* mice are labelled in blue and the most discriminatory increases in *Hexb ^−/−^* mice are labelled in red. (**B**) concentration of *N*-acetylaspartate in spinal cord tissue from one to four-month-old control and *Hexb ^−/−^* mice (two resonances shown). (**C**) concentration of myo-inositol in spinal cord tissue from one to four-month-old control and *Hexb ^−/−^* mice. (**D**) expanded region (1.96–2.095 ppm) from the ^1^H-NMR spectra of cerebrum tissue from four-month-old control *Hexb ^−/−^* mice. The resonance corresponding to *N*-acetylaspartate is decreased in the *Hexb ^−/−^* mouse. (**E**) concentration of leucine and valine in cerebrum tissue from one-month-old control and *Hexb ^−/−^* mice. (**F**) concentration of leucine and valine in spinal cord tissue from one-month-old control and *Hexb ^−/−^* mice. (**G**) concentration of three neurotransmitters, aspartate, glutamate and GABA, in cerebrum tissue from *Hexb ^−/−^* mice. Control concentration is set to 1. (**H**) concentration of aspartate, glutamate and GABA, in spinal cord tissue from control and *Hexb ^−/−^* mice. Control concentration is set to 1. Results are presented as mean +/− SEM. Significance level quoted for Student’s *t*-test * *p* ≤ 0.05; ** *p* ≤ 0.01; *** *p* ≤ 0.001.

**Figure 6 metabolites-11-00018-f006:**
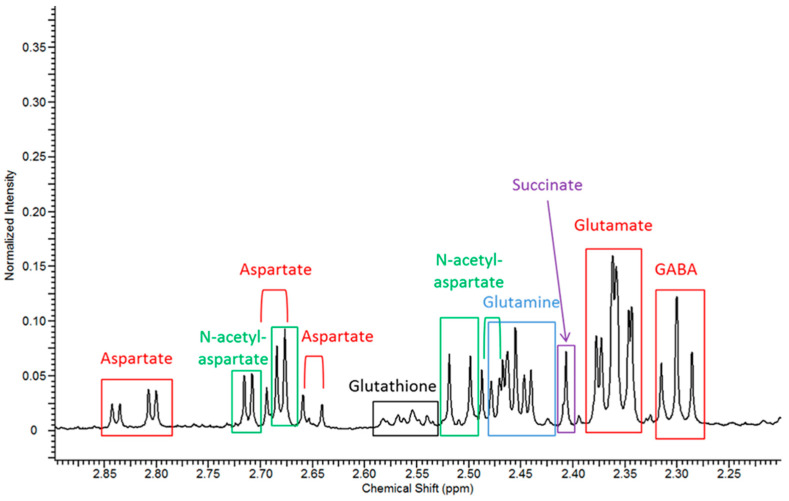
Expanded region (2.20–2.90 ppm) from the ^1^H-NMR spectrum of cerebrum tissue from a four-month-old control mouse showing the presence of resonances for three neurotransmitters (labelled in red), GABA, glutamate and aspartate, as well as succinate, glutamine, glutathione and the neuro-axonal marker *N*-acetylaspartate.

**Figure 7 metabolites-11-00018-f007:**
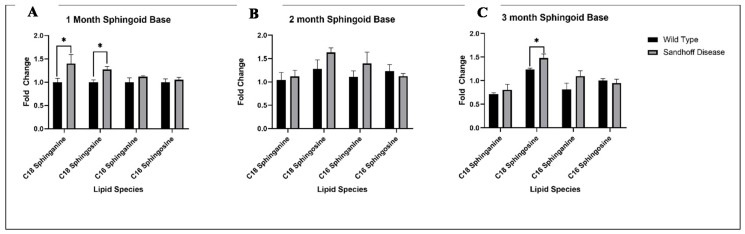
Fold changes of sphingoid bases between wild type and Sandhoff disease mice, identified in the inside out sphingolipidomics aqueous fraction using reverse phase LC-MS. Fold changes between wild type and SD mouse model for (**A**) 1 month, (**B**) 2 months and (**C**) 3 months. Analysed by Student *t*-test, WT: n = 5, SD: n = 5, *p* < 0.05 *.

**Figure 8 metabolites-11-00018-f008:**
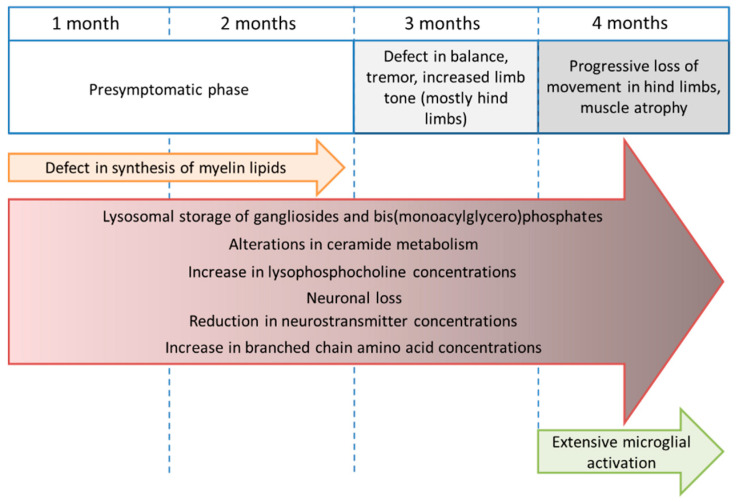
Representation of the metabolic changes observed in the metabolomic study of a mouse model of SD from one to four months of age, matched with the manifestations observed in this murine model.

**Table 1 metabolites-11-00018-t001:** Parameters used for the targeted analysis by mass spectrometry of ceramides reported in the paper.

Compound	MRM (Da)	Dwell Time (ms)	Declustering Potential (V)	Collision Energy (V)	Collision Cell Exit Energy (V)
C14-Cer	510.523 → 264.300	150	71	39	6
C16-Cer	538.479 → 520.500	150	46	17	14
C18-Cer	566.402 → 548.700	150	36	19	48
C18:1-Cer	564.374 → 173.200	150	61	13	14
C20-Cer	594.483 → 576.500	150	96	23	8
C22-Cer	622.636 → 604.600	150	81	19	18

## Data Availability

Data is available through the MetaboLights repository (www.ebi.ac.uk/metabolights/).

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
