# Peer review of "Decrease in Myelin-Associated Lipids Precedes Neuronal Loss and Glial Activation in the CNS of the Sandhoff Mouse as Determined by Metabolomics"

_metabolites, 2020, doi:10.3390/metabo11010018_

Round 1
Reviewer 1 Report
This is an article by Lecommandeur et al., describing a decrease in myelin-associated lipids that precedes neuronal loss and glial activation in the CNS of Sandhoff mice determined by metabolomics. They authors measured global changes in neuronal and myelin lipids and other metabolites over the course of Sandhoff disease in mice. They found that myelin lipids are decreased at one month of age, whereas evidence for neuronal loss (decrease of neuronal metabolites like N-acetyl aspartate, glutamate and GABA and increase in LPC) are detected and later in the course of the disease. Also, myo-inositol increase (indicator of microglia activation) correlates with the later stages of the disease.
This is a very careful and detailed analysis of neuronal and myelin metabolites by measuring changes during the progression of Sandhoff disease in brain and spinal cord.
I do not have any major comments for this work. The conclusions are supported by the results shown. The article is well written, figures are clear.
Minor comment:
Line 121-122 and Figure 3: The title of figure 3 is “Concentration of two LPC species in cerebrum and spinal cord tissues…” and the graphs show LPC 18:0, 20:4 and 22:6. Also, in line 121-2, authors mentioned LPC 16:0 which is increased at 2 months and LPC 18:01 is increased at 2 and 3 months. Are these data not shown?
Author Response
We thank reviewer 1 for the overall complimentary and supportive comments concerning our manuscript. We also thank the reviewer for spotting the typographical errors which we have now corrected as detailed below.
- Line 121-122 and Figure 3: The title of figure 3 is “Concentration of two LPC species in cerebrum and spinal cord tissues…” and the graphs show LPC 18:0, 20:4 and 22:6. We have corrected the figure legend so it now says three lyso-phosphatidylcholines for the figure legend to be consistent with the actual figure itself.
- Also, in line 121-2, authors mentioned LPC 16:0 which is increased at 2 months and LPC 18:01 is increased at 2 and 3 months. Are these data not shown? This data is not shown in the manuscript and we have now included the comment "data not shown" at the end of the sentence.
Reviewer 2 Report
General comment:
This paper studied a Sandhoff disease mouse model at the metabolomic level. The authors provide a wide array of data hinting to neuronal degeneration and myelination issue over the progression of the disease in this mouse model through the fluctuation of several lipids and neurotransmitters over the time course. While mass spectrometry was used for the analysis of the organic phase it was not used for the aqueous phase which is confusing since some of the short coming mention at the bottom of page 11 could have been solved by simply running positive mode analysis of the aqueous phase. Not with standing, the data provided and the integration of previous work support well the authors hypothesis and further research might lead to the development of treatment for these type of orphan diseases.
Major Comments:
- The authors never provide an explanation as to why the cerebrum and spinal cord are the only two part of the nervous system, they decided to analyze. A few sentences on their significance over other part of the nervous system would help understand the work at hand and its significance.
- Figure 1 the authors present the data of the controls against the four-month-old cohort. Why is their no data for month 1 through 3? These could easily be added as supplemental figures.
- Figure 2 the authors present H-NMR data and label half resolved peak as actual molecule without referencing any H-NMR library or providing H-NMR from standard to support their annotation. H-NMR of complex mixture as a way of detection is a weird choice when having access to mass spectrometers.
- Page 5 last paragraph. The authors identify GalCer through exact mass and 1 MS/MS transition. It is quite well known in mass spectrometry that we cannot differentiate GluCer from GalCer through exact mass and that MS/MS transition. The authors should provide a quick statement as to why we should believe these are GalCer or simply label them HexCer as most people do in the literature.
- Furthermore, when annotating HexCer the authors should provide information about their oxidation state. Are they di- or tri- oxidized? HexCer(d18:1/16:0) is quite different from HexCer(t18:1/16:0).
Minor Comments:
- On page 2, the authors mention that the GM2/GA2 accumulation is detectable in all region of the CNS. Other lysosomal storage disorders have been shown to have some rather specific accumulation pattern throughout the brain. Is this the case for SD to or is this disease an all hand on deck situation?
- On page 2 around line 76-80, the authors mention the increase in BMP(22:6/22:6) being more stark than other BMP. Can the authors comment on the possible importance of the two DHA side chain and their possible role in the disease?
- The way the authors annotate plasmalogen-PE is not standard. PE (p36:1) would be the right way to do it. Also please put bracket in Figure 4 around the plasmalogen carbon and unsaturation count like it’s done for HexCer.
- Figure 4. Can the authors provide an explanation for the lack of significance at the 2nd month when there is significance for at least one which is still significant at the 3rd month? Also where is the 4th month data for this?
- Since the data point to myelination issue, have the authors look at cholesterol changes in their dataset?
Author Response
We thank the reviewer for their detailed comments and their suggestions for the improvement of the paper. We have acted on all the comments raised by the reviewer and provide a detailed response below.
Major comments
1. The authors never provide an explanation as to why the cerebrum and spinal cord are the only two part of the nervous system, they decided to analyze. A few sentences on their significance over other part of the nervous system would help understand the work at hand and its significance.
GM2 affects all the CNS but the initial microscopic neuropathology is in the cerebrum which is neurone rich in the cortex (grey matter), and also clinically associated with the rapid dementia. On the other hand, with subcortical axons, the spinal cord is rich in myelinated axons. Thus, these major structures typify the range of ‘grey - white’ tissue divisions of the CNS studied in this disease which is why we have compare the two regions. We have included this information in the manuscript.
2. Figure 1 the authors present the data of the controls against the four-month-old cohort. Why is their no data for month 1 through 3? These could easily be added as supplemental figures.
We have added data for some of the key ganglioside changes across the first 3 months to accompany the results presented in figure 1 to demonstrate that these changes do indeed occur across the four months.
3. Figure 2 the authors present H-NMR data and label half resolved peak as actual molecule without referencing any H-NMR library or providing H-NMR from standard to support their annotation. H-NMR of complex mixture as a way of detection is a weird choice when having access to mass spectrometers.
The assignment of metabolites by NMR spectroscopy is discussed in the methods section on page 16, line 418. Perhaps for those less familiar with NMR spectroscopy, once performed on a similar tissue the approach is relatively straight because of chemical shift and coupling patterns which provide rich structural information for small molecules. Also we have used NMR spectroscopy because of the wealth of information on 1H Magnetic Resonance Spectroscopy changes in the brain during various neurological disorders which we can discuss. We have included comment on this in the manuscript (page 5, line 109).
4. Page 5 last paragraph. The authors identify GalCer through exact mass and 1 MS/MS transition. It is quite well known in mass spectrometry that we cannot differentiate GluCer from GalCer through exact mass and that MS/MS transition. The authors should provide a quick statement as to why we should believe these are GalCer or simply label them HexCer as most people do in the literature.
We thank the reviewer for this excellent comment which they are quite right about. We were assuming the changes were largely GalCer because of its rich content in brain tissue but we cannot definitively say this on our data. We have relabelled the changes as HexCer as discussed and included a brief explanation in the manuscript.
5. Furthermore, when annotating HexCer the authors should provide information about their oxidation state. Are they di- or tri- oxidized? HexCer(d18:1/16:0) is quite different from HexCer(t18:1/16:0).
Again, the reviewer makes an excellent point and we have adjusted our assignments accordingly.
Minor comments:
1. On page 2, the authors mention that the GM2/GA2 accumulation is detectable in all region of the CNS. Other lysosomal storage disorders have been shown to have some rather specific accumulation pattern throughout the brain. Is this the case for SD to or is this disease an all hand on deck situation? This is an excellent question by the reviewer. As we explain in point 1 in Sandhoff disease the metabolic changes are widespread. Furthermore, we have shown that this is found in a number of lysosomal storage disorders (both in the brain and systemically) and perhaps the question should be why certain regions are predisposed to pathology rather than the metabolic deficits per se.
2. On page 2 around line 76-80, the authors mention the increase in BMP(22:6/22:6) being more stark than other BMP. Can the authors comment on the possible importance of the two DHA side chain and their possible role in the disease? This is an interesting question. The brain is particularly rich in DHA and in turn BMPs rich in 22:6 would be expected to accumulate in the lysosomes of the brain. We have included this comment in the manuscript (page 12, line 265).
3. The way the authors annotate plasmalogen-PE is not standard. PE (p36:1) would be the right way to do it. Also please put bracket in Figure 4 around the plasmalogen carbon and unsaturation count like it’s done for HexCer. This is a good point and we have changed are terminology in figure 4 and the results section.
4. Figure 4. Can the authors provide an explanation for the lack of significance at the 2nd month when there is significance for at least one which is still significant at the 3rd month? Also where is the 4th month data for this? We have presented the 1-4 month data for one plasmalogen-phosphatidylethanolamine in panels B and C. We are unsure why only the early time point was significant but it is possible that there are some adaptive changed that occur in the brain at later time points. It's also possible that we were just not powered enough to detect changes at each time point.
5. Since the data point to myelination issue, have the authors look at cholesterol changes in their dataset? We did detect cholesterol and cholesterol esters in our datasets but these were not highlighted as major changes in the multivariate statistics (unlike the liver where the cholesterol changes were much more marked).